# Oncogenic Linear Collagen VI of Invasive Breast Cancer Is Induced by CCL5

**DOI:** 10.3390/jcm9040991

**Published:** 2020-04-02

**Authors:** Elizabeth Brett, Matthias Sauter, Éadaoin Timmins, Omid Azimzadeh, Michael Rosemann, Juliane Merl-Pham, Stefanie M. Hauck, Peter J. Nelson, Karl Friedrich Becker, Ilse Schunn, Aoife Lowery, Michael J. Kerin, Michael Atkinson, Achim Krüger, Hans-Günther Machens, Dominik Duscher

**Affiliations:** 1Department of Plastic and Hand Surgery, Technical University Munich, Ismaninger Strasse 22, 81675 Munich, Germany; eliza.brett@tum.de (E.B.); matthias.sauter@tum.de (M.S.); hans-Guenther.Machens@mri.tum.de (H.-G.M.); 2Department of Anatomy, National University of Ireland, Galway, Newcastle Road, H91 YR71 Galway, Ireland; eadaoin.timmins@nuigalway.ie; 3Helmholtz Zentrum München, Institute of Radiation Biology, Ingolstaedter Landstraße 1, 85764 Neuherberg, Germany; omid.azimzadeh@helmholtz-muenchen.de (O.A.); rosemann@helmholtz-muenchen.de (M.R.); atkinson@helmholtz-muenchen.de (M.A.); 4Helmholtz Zentrum München (GmbH), Research Unit Protein Science, Heidemannstr. 1, 80939 München, Germany; juliane.merl@helmholtz-muenchen.de (J.M.-P.); hauck@helmholtz-muenchen.de (S.M.H.); 5Research Group Clinical Biochemistry, Nephrology Center, Department of Internal Medicine IV, Klinikum der Universität München, Schillerstr. 42, 80336 Munich, Germany; peter.nelson@med.uni-muenchen.de; 6Institute of Pathology, Technische Universität München, Trogerstr. 18, 81675 Munich, Germany; kf.becker@tum.de; 7Technical University of Munich, Medical Materials and Implants, Boltzmannstr. 15, 85748 Garching, Germany; ilse.schunn@tum.de; 8Discipline of Surgery, Lambe Institute for Translational Research, Newcastle Road, H91 YR71 Galway, Ireland; aoife.lowery@nuigalway.ie (A.L.); michael.kerin@nuigalway.ie (M.J.K.); 9Institute of Molecular Immunology and Experimental Oncology, Technical University Munich, Ismaninger Strasse 22, 81675 Munich, Germany; achim.krueger@tum.de

**Keywords:** triple negative breast cancer, linear, border, collagen VI, CCL5, invasion, metastatic, adipose derived stem cell, extracellular matrix, decellularization

## Abstract

The triple-negative breast tumor boundary is made of aligned, linear collagen. The pro-oncogenic impact of linear collagen is well established; however, its mechanism of formation is unknown. An in vitro analogue of the tumor border is created by a co-culture of MDA-MB-231 cells, adipose derived stem cells, and dermal fibroblasts. Decellularization of this co-culture after seven days reveals an extracellular matrix that is linear in fashion, high in pro-oncogenic collagen type VI, and able to promote invasion of reseeded cells. Further investigation revealed linear collagen VI is produced by fibroblasts in response to a paracrine co-culture of adipose derived stem cells and MDA-MB-231, which together secrete high levels of the chemokine CCL5. The addition of monoclonal antibody against CCL5 to the co-culture results in an unorganized matrix with dramatically decreased collagen VI. Importantly, reseeded cells do not exhibit pro-oncogenic behavior. These data illustrate a cellular mechanism, which creates linear extracellular matrix (ECM) in vitro, and highlight a potential role of CCL5 for building striated tumor collagen in vivo.

## 1. Introduction

Triple negative breast cancer (TNBC) fails to express receptors for progesterone, estrogen, and HER2 (PR-/ER-/HER2-) [1] and represents the most aggressive form of breast cancer [2]. TNBC tumors exhibit an outer layer of extracellular matrix (ECM) featuring linear arrays of collagen radiating 90° perpendicular from the tumor border [3]. These collagenous structures serve to aid intravasation, expand the range of the primary tumor, and extend through breast parenchyma [4,5]. However, the deposition mechanics of the linearized matrix are totally unknown. 

CCL5 (C-C motif chemokine ligand 5) is a member of the C-C chemokine family, and acts as a potent chemoattractant of T cells, basophils, eosinophils, and macrophages [6]. It has been shown that the main producer of tumor associated CCL5 is not the cancer cells, rather the local mesenchymal stem cell population [7,8]. The literature is unanimously agreed on the pro-oncogenic impact of CCL5 on invasive breast tumors [6,9]. The presence of CCL5 in biopsy samples has been reported as a prognostic indicator for stage II breast cancer progression [10]. Similarly, circulating CCL5 was seen to increase with the number of involved lymph nodes in serum samples from invasive breast cancer patients [11,12,13]. As an intervention, monoclonal antibody injected intraperitoneally against CCL5 showed a reduction in mesenchymal stem cell-induced breast tumor metastasis in a mouse model [8]. 

In vitro extracellular matrix (ECM) deposition studies have been used to test the influence of ECM produced by aggressive melanoma on the epigenetic conversion of less aggressive melanoma cells [14]. Considering the unique striated matrix of the TNBC tumor, and its reported role in metastasis promotion, we adapted this in vitro model to recapitulate the cellular interplay at the tumor/host border. We also used this tumor border analogue to investigate cell types and molecular inducers responsible for creating pro-oncogenic breast cancer matrix.

## 2. Materials and Methods

### 2.1. Experimental Design 

The objective of this study was twofold: to apply established in vitro protocols to a novel, untested hypothesis, and to examine the subsequent findings on invasive tumor extracellular matrix. The co-culture groups were selected with special attention to creating an analogue of the invasive tumor border, while allowing for experimental control. The pathway of this work was defined by results from unbiased experimental design; hence, the explorative tone of the study.

### 2.2. Cell Line Acquisition, Culture, and Co-Culture Arrangement

Human fibroblasts (HS-27), adipose derived stem cells (ASC), and TNBC cells (MDA-MB-231) were all purchased as cell lines from Lonza. Cultures of individual cell types were carried out under normal conditions (37 °C, 5% CO_2_). HS-27 and MDA-MB-231 were cultured in Dulbecco’s modified Eagle’s medium, 10% fetal bovine serum, 1% PenStrep (Gibco). ASCs were cultured in StemMACS media (Miltenyi Biotech), with 1% PenStrep. Co-cultures were arranged per Figure 1A. Group (i) (cancer juxtacrine group) featured all three cell types combined, in equal number (9.6 cm^2^ = 50,000 total cells). Group (ii) (cancer paracrine group) had 25,000 HS-27 and 25,000 ASCs per 9.6 cm^2^ and an overlying boyden chamber containing 20,000 MDA-MB-231. Group (iii) (healthy group) had 25,000 HS-27 and 25,000 ASCs per 9.6 cm^2^.

### 2.3. Matrix Deposition and Decellularization 

Groups were arranged as described in Figure 1A and cultured per the workflow in Appendix A. Briefly, cultures were grown for 7 days. Every other day, media was changed to contain 50 mM ascorbic acid. In the experiments blocking and supplying CCL5, a monoclonal antibody against human CCL5 (kindly supplied by P.J.N., L.M.U.), or recombinant human CCL5 (Peprotech) was added daily to the media for a final concentration of 10 μg/mL^−1^. On day eight of culture, plates were washed 2 × phosphate buffered saline (PBS), and decellularized using a 20 mM NH_4_OH/1% Triton-X solution, at 37 °C for four minutes. The decellularization solution was diluted by twice its volume of PBS containing 1% PenStrep, and then stored at 4 °C overnight. The following day, plates were washed 3 × PBS, and prepared suitably for experimentation.

### 2.4. ECM Quantification

Once matrices were washed 3 × PBS, protein was quantified using a Pierce^TM^ BCA Protein Assay Kit (Thermo Fisher Scientific, Waltham, MA, USA), per manufacturer’s protocol.

### 2.5. Cytokine Array 

A pooled total of 1 ml media was taken from 3 replicates of each group on day five of the culture. Pooled supernatant was run on a Human Cytokine Array C5 (RayBiotech, Norcross, GA, USA) per manufacturer’s protocol. Arrays were imaged on a ChemiDoc Imaging System (Bio-Rad, Hercules, CA, USA) at an exposure time of one minute.

### 2.6. Atomic Force Microscopy and Scanning Electron Microscopy

Samples of decellularized ECM were washed with PBS and air-dried for 48 h. Using Dimension^TM^ 3100 Atomic Force Microscope (Bruker/Digital Instruments, Santa Barbara, CA USA), matrices were measured by POINTPROBE PLUS^®^ (silicon tips, resistivity 0.01–0.02 Ω, PPP-NCHR-10). Images were analyzed using Gwyddion software (V2.55, Brno, Czech Republic). Roughness was analyzed using ‘Roughness Calculation’ plugin for ImageJ (2002). Sample preparation for SEM involved identical decellularization and washing, followed by fixation in 3.5% formaldehyde, and dehydration in grades of alcohol from 40%, 60%, 70%, 80%, 90%, 95%, and 100% ethanol. Samples were air-dried overnight, then gold sputter coated and imaged by Jeol JSM-6390 scanning electron microscope at 10,000× magnification.

### 2.7. Histology and Immunohistochemistry 

Biopsies of human TNBC tumor border were taken from 3 patients (per Technical University of Munich guidelines, ethics vote #2997/10) were prepared as paraffin embedded blocks and cut at a thickness of 70 µm. Hematoxylin and eosin staining was performed as previously described [15]. Immunohistochemical staining was performed using the manufacturer’s protocol, with anti-human CCL5 (raised in goat), FITC (fluorescein isothiocyanate) conjugated donkey anti-goat and anti-human collagen VI (raised in mouse), Alexa Fluor 555 conjugated goat anti-mouse (Abcam). Fluorescent images were taken using a Zeiss ObserverZ.1 (AX10, Jena, Germany) and analyzed using ImageJ (NIH).

### 2.8. Cell Reseeding and Imaging 

Decellularized ECM was washed 3 × PBS, followed by fully supplemented media containing 20,000 MDA-MB-231 per 9.6 cm^2^. Cells were imaged under Brightfield microscopy every 24 h until completely confluent.

### 2.9. Cell/Matrix Preparation for Mass Spectroscopy

Co-cultures for groups (i) and (iii) were seeded and cultured under normal conditions using the cell quantities described above. Control arm: 10,000 MDA-MB-231 cells per 9.6 cm^2^ were reseeded onto the matrices and cultured for 24 h. The cell/matrix extracts were then solubilized in RIPA (radioimmunoprecipitation) buffer containing protease inhibitor and processed for mass spectroscopic analyses. The cell/matrix extracts were then solubilized in RIPA buffer containing protease inhibitor and processed for mass spectroscopic analyses.

### 2.10. Mass Spectrometry and Proteomic Analysis

Filter aided sample preparation (FASP) digest was performed on 10 µg of cell/matrix extracts, which were digested using a modified FASP procedure, as previously described [16]. LC-MSMS measurements were made according to methods previously described (for details see Appendix A). Quantitative data analysis was performed using Progenesis QI for proteomics [17]. Gene ontology term analysis (molecular function) was performed using Ingenuity Pathway Analysis (IPA, Qiagen) (QIAGEInc., https://www.qiagenbioinformatics.com/products/ingenuity-pathway-analysis) [18].

The analyses of protein-protein interaction were performed by the software tools STRING 11 (http://string-db.org) [19].

### 2.11. Qualitative Real Time PCR

RNA extraction was performed using TRIzol™ agent (Thermo Fischer Scientific, Waltham, MA, USA) and RNeasy Minikit (Qiagen, Hilden, Germany). Reverse transcription and PCR were using a OneStep RT-PCR.

Kit (Qiagen, Hilden, Germany). Primers for CCL5 from Invitrogen (Forward: GTGGCAGGCAGTAAGATAAACTTG, reverse: CAAAAAGCTTCCCCAACTAAAGC).

### 2.12. Statistical Analysis

Statistical analyses were performed with GraphPad Prism software (GraphPad Software Inc., San Diego, CA, USA). Data are presented as the mean ± standard deviation. Two-tailed *t*-tests were run to directly compare two groups together, and ANOVA was used for multiple group comparisons. *p* value of <0.05 was considered significant. 

## 3. Results 

### 3.1. Extracellular Matrix Formed under Juxtacrine Influence has Smooth, Striated Ultrastructure 

In order to evaluate matrix deposition, 3 co-culture groups (per Appendix A) were seeded, cultured for seven days, and decellularized. There were grossly visible differences between the matrices generated (Figure 1A). ECM formed by the juxtacrine cancer co-culture (group (i): adipose derived stem cells (ASCs), fibroblasts, MDA-MB-231 cultured in juxtacrine) had a glassy, smooth appearance, with a linear, striated structure. In agreement with the literature, linear collagen has been identified histologically in both canine and human breast tumor samples, specifically at the tumor border [4,5]. In contrast to group (i), ECM formed by the paracrine co-culture (group (ii): ASCs and fibroblasts cultured in juxtacrine, MDA-MB-231 cultured in paracrine, via Boyden chamber) was sparse, with disorganized fibrils laid down in no discernable pattern. In the cancer cell-free juxtacrine co-culture (group (iii): ASCs and fibroblasts cultured in juxtacrine), the ECM formed a rich network of collagen in a mesh formation (Figure 1A). Atomic force microscopy (AFM) confirmed the differing three-dimensional structures of ECM from groups (i), (ii), and (iii). AFM micrographs show the linear ECM generated by group (i) (maximum height 0.11 µm), the disorganized minimal ECM of group (ii) (maximum height 80 nm), and a thick meshwork of ECM from group (iii) (maximum height 0.13 µm) (Figure 1B). Quantification showed similar amounts of ECM secreted between groups (i) and (iii), and over 50% less ECM deposited in group (ii). (Figure 1C). Surface area calculation showed the matrix formed by group (iii) was the roughest, with a significant difference compared to (i) and (ii) (*p* = 0.008, Figure 1D). Cultures of only MDA-MB-231 under paracrine influence of either fibroblast or ASC did not produce quantifiable matrix (Appendix A), and reseeded MDA-MB-231 formed a uniform monolayer, without exhibiting enhanced invasive behavior (Appendix A). Only the co-culture of ASCs, fibroblasts, and MDA-MB-231 formed a linear matrix seen by SEM and induced reseeded cells into road-like structures (Appendix A). Given these significant morphological differences and ability to instruct reseeded MDA-MB-231 cells, the next stage was to test the protein make-up of the matrices produced.

### 3.2. Striated Collagen VI Found in Tumor Border and Juxtacrine Cancer Matrix 

Proteomics analysis of reseeded MDA-MB-231 and matrix combined revealed a significantly higher proportion of collagen VIα3 (yellow data point, Figure 2A present in group (i) compared to group (iii). Congruently, it has been reported that the invasive front of the TNBC tumor is made chiefly of collagen VI [20,21]. The C-terminal breakdown product of the COL6α3 chain (C5) is endotrophin; a pro-fibrotic, pro-angiogenic, pro-EMT (epithelial-mesenchymal transition), chemoattractant, which has been seen to decrease the efficacy of cisplatin treatment [22,23]. In this way, collagen VI can be thought of as a structural and signaling molecule. H&E stains of human biopsy samples of the tumor border showed long, well-formed extensions from the tumor into the parenchyma, shown histologically and schematically in Figure 2B. Immunohistochemistry on human biopsy samples at the tumor border revealed collagenous structures positively stained for collagen VI, and CCL5 (Figure 2C). MDA-MB-231 cells reseeded onto acellular ECM in vitro grew in road-like patterns, observed only on the ECM of group (i) (juxtacrine cancer co-culture). Cells grew in a monolayer on groups (ii) and (iii), as normal cell cultures appear (Figure 2D). We observed organized structures forming five days after reseeding on matrix from group (i), with some contiguous tracks spanning approximately 360–390 µm (Figure 2E). While it is likely the reseeded cells are doing no more than following the physical three-dimensional (3D) linear collagen, it illustrates the unique physical relationship between cancer generated matrix and cancer cell behavior. Gene ontology term analysis on proteome of cells reseeded on the matrix featured ‘cell cycle’,‘cell movement’, and ‘post-translational modification’ as the top three enriched terms (Figure 2F). Collectively, signaling protein analysis revealed a highly activated pathway of post-translational modification and protein transport to the rough endoplasmic reticulum (via GFM, MRPS12, RPS21, and SRP9), suggested circuitry to predict breast cancer outcomes [24] (Figure 2G). Considering the high collagen VI level in the cancer juxtacrine matrix, it was important to understand, next, the paracrine signaling occurring among the cells of group (i) compared to group (ii) as matrices were being deposited. 

### 3.3. Blocking CCL5 in Juxtacrine Co-Culture Disrupts the Linear Deposition of ECM 

A cytokine screen of the matrix deposition co-cultures (Figure 3A) on day five revealed an increase of CCL5 only in group (i) (Figure 3B). Increased levels of CCL5 through juxtacrine contact of TNBC cells with healthy stem cells has been described previously. Similar to our findings, CCL5 was not upregulated with cancer cells in a paracrine relationship to non-cancer cells [8]. This establishes a unique cytokine profile at the juxtacrine level and is especially interesting to observe the same co-culture under CCL5-depleted conditions. By addition of a monoclonal antibody against CCL5 during the matrix deposition period in group (i), there was an increase of phase contrast under Brightfield microscopy of the cultured cells. A live/dead assay showed 100% cell viability coupled with visible isolation of the cell types from one another (Figure 3C). CCL5 is known to enhance binding of leukocytes and T cells via CD44 signaling, an adhesion molecule common in many cancers [25,26]. These data infer a relationship where cellular juxtacrine binding is CCL5-mediated. Cells were cultured for the eight-day deposition protocol, then ECM was decellularized (as in Appendix A). AFM and SEM show that blocking CCL5 in vitro results in a disrupted, unorganized, homogenous layer of ECM, compared to its normal linear format when CCL5 is unlimited (Figure 3D). Maraviroc is a competitive CCR5 antagonist, given systemically as HIV treatment to combat CCL5-mediated CD44 signaling in T-cells [27]. An in-silico test was run to demonstrate the reductive effect of Maraviroc on breast tumor growth rates [28], as a concept study for secondary drug application. From a cytokine perspective, Maraviroc may represent an alternative, non-hormonal drug treatment, which could specifically act to limit the physical construction of the linear matrix. The next important effect of blocking CCL5 to be tested is whether it affects the proportion of collagen VI in the matrix of group (i), and if there is any difference in the reseeded cellular behavior on matrix formed in CCL5^low^ conditions.

### 3.4. Blocking CCL5 in Juxtacrine Co-Culture Reduces Collagen VI and Reseeded Cell Oncogenic Behavior 

Addition of recombinant CCL5 to the healthy and cancer juxtacrine cocultures did not reveal an altered ECM morphology; matrices from group (i) remain linear and stain positively for collagen VI. Matrices from group iii retain their meshwork pattern (Figure 4A). While addition of recombinant CCL5 to group (i) did not significantly alter the levels of ECM deposited, or the proportion of collagen VI therein, blocking CCL5 served to drastically decrease both the amount of total ECM deposited and collagen VI (* *p* = 0.041, ** *p* = 0.0049) (Figure 4B). Linked to in vivo literature, a CCR5^−/−^ mouse model on the background of the spontaneous mammary carcinoma model (mammary tumor virus—polyoma middle T antigen (MMTV-PyMT)), significantly delayed the crucial early stages of invasive tumor development [29]. Reseeded MDA-MB-231 cells on the matrix formed with a CCL5 blocker exhibited low positive staining of integrin β1, and a normal monolayer culture without road-like structures. This finding differs to the cells reseeded on the matrix formed under unlimited CCL5; which formed road-like structures and showed higher positive staining for integrin β1 (Figure 4C), significantly different after three days of culture (* *p* = 0.029, Figure 4D). Integrin β1 is a known driver of TNBC metastasis, and a clinical prognostic of TNBC development [30], by promoting invasion, proliferation, and cell survival. It so follows that silencing integrin β1 in vitro limits the oncogenic ability of MDA-MB-231 cells [31]. The next step in this study was to understand the origin of CCL5, and the main producer of linear matrix.

### 3.5. Fibroblasts Under Paracrine Influence of ASC/MDA-MB-231 Produce Linear Collagen VI Matrix 

While matrix produced by fibroblasts alone under the paracrine influence of fibroblast/MDA-MB-231 stains positively for collagen VI, it is irregular and in a random meshwork pattern. However, fibroblasts under the paracrine influence of ASC/MDA-MB-231 produce linear collagen VI matrix as seen under fluorescent microscopy (Figure 5A). qPCR reveals significantly increased expression of CCL5 in ASC/MDA-MB-231 co-culture, compared to fibroblast/MDA-MB-231 co-culture. Congruently, conditioned media taken during the matrix deposition period shows higher CCL5 presence in ASC/MDA-MB-231 media compared to fibroblast/MDA-MB-231 media. These data highlight a previously known finding; stem cell membranes bound to MDA-MB-231 cell membranes create CCL5 [8]. Taken collectively, there is evidence for an axis involving ASC/MDA-MB-231 signaling that produces CCL5, which in turn stimulates fibroblasts to produce striated collagen VI (Figure 5D).

## 4. Discussion 

At the boundary of the primary tumor, there is membrane-membrane contact of cancer cells with the neighboring non-cancer cell populations. We hypothesize that the ASCs at the TNBC tumor boundary bind to MDA-MB-231 cells, generating high levels of CCL5. The resident fibroblasts react to CCL5 by forming linearized ECM. Our data show the striated ECM generated by in vitro co-culture is rich in collagen type VI, and able to promote integrin β1 expression of reseeded cells. Moreover, the linear ultrastructure is dependent on CCL5, which the co-culture of ASC/MDA-MB-231 generates. Specifically, this study aims to connect the existing TNBC research on CCL5 [8], linearized matrix [3], collagen VI [20], and invasive cellular behavior [31]. 

The finding of linear collagen at the breast tumor border echoes multiple TNBC publications. ‘Tumor associated collagen signatures’ (TACS) was first coined in 2006 [3], and represents a system of distinct layers of collagen, which radiate out from an invasive breast tumor. TACS1 defines the innermost layer, a densely packed shell of collagen. TACS2 represents the middle zone, made of collagen aligned in onion-like spheres around the tumor. TACS3 being the outermost, features branches of collagen diverging 90° perpendicular to the tumor boundary [3]. We present evidence that these linear structures are made of collagen type VI, a collagen isoform whose pro-oncogenic impact is recapitulated in vivo. Col6a1^−/−^ mice on a background of MMTV-PyMT show decelerated hyperplasia and primary tumor formation [32]. 

The present study is limited by the two-dimensional culture model and use of only triple negative breast cancer cells. Similarly, the intervention is not in vivo and there is no pharmacological investigation. However, the limitations of this study represent future research focuses. These findings, when extrapolated to the clinic, could inform diagnostics, as well as medical or surgical treatment of TNBC. A pivoted role of the anti-CCL5 HIV drug Maraviroc could feasibly integrate into TNBC treatment. Its potential has been identified and even patented for TNBC therapy since 2012 (patent #CA2873743A1). Investigation into CCL5 and breast density could inform radiologists on difficult imaging cases, considering the desmoplastic milieu created by radiotherapy [33]. 

There is a known linearized matrix in breast cancer, which we have seen in vitro is formed chiefly by fibroblasts under CCL5 signaling. The striated structure of the matrix allows for oncogenic cell behavior and is disrupted by blocking CCL5 in vitro during matrix deposition. This represents an unreported role of CCL5 in specific ECM construction, which stands to illuminate ways of surgically excising, diagnosing, and treating invasive breast cancer. 

## Figures and Tables

**Figure 1 jcm-09-00991-f001:**
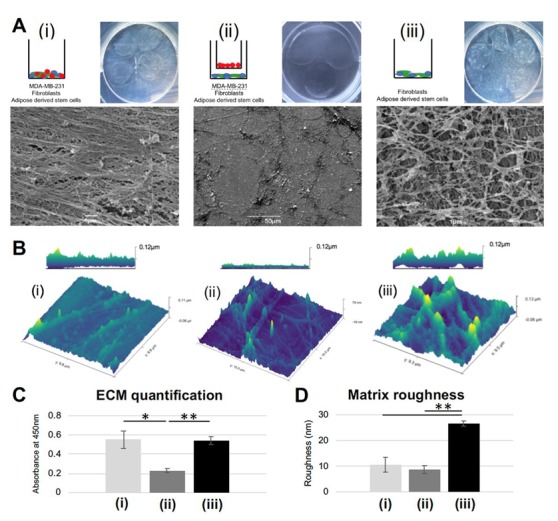
Striated extracellular matrix (ECM) results only from a juxtacrine co-culture of MDA-MB-231, adipose derived stem cells (ASC), and fibroblasts: (**A**) Gross images and SEM micrographs show the decellularized layer of ECM deposited by the schematic matrix: (**i**) juxtacrine cancer group; (**ii**) paracrine cancer group; (**iii**) juxtacrine healthy group. (**B**) Atomic force microscopy (AFM) images for the decellularized layers from groups i, ii, and iii. (**C**) Quantification of deposited ECM. Error bars represent SD. (* *p* <0.05, ** *p* <0.01). (**D**) Quantification of roughness per AFM images. Error bars represent SD. (** *p* <0.01).

**Figure 2 jcm-09-00991-f002:**
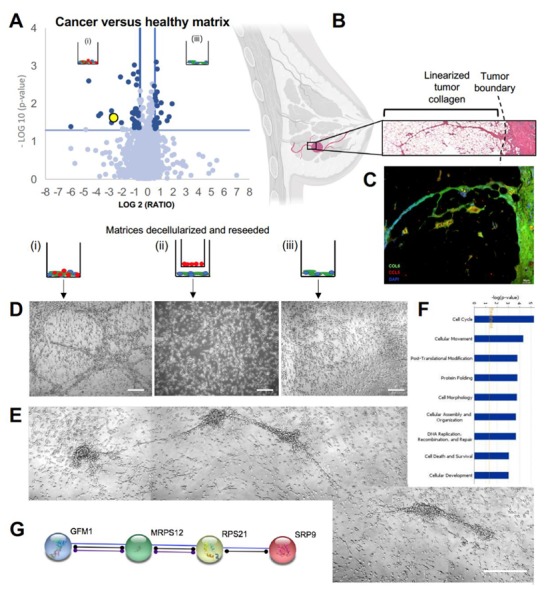
Collagen VI found in striated in vitro matrix and human tumor border biopsies: (**A**) volcano plot depicting proteomic differences between the cancer juxtacrine and healthy juxtacrine matrices with reseeded cells. Yellow datapoint represents COL6A3. (**B**) Schematic of invasive breast tumor accessing tissue planes via linear collagen, and a blown-up hematoxylin and eosin image of the tumor boundary. Breast image created with biorender.com. (**C**) Immunohistochemistry of breast tumor border. Collagen type 6 stained in green, CCL5 stained in red, DAPI stained in blue. Scale bar 50 μm. (**D**) Trio of Brightfield images show reseeded cell morphology on each of the three decellularized ECM. Scale bar 100 μm. (**E**) Isolated example of road structures formed by the reseeded cells on matrix from group i (cancer juxtacrine). Scale bar 100 μm. (**F**) Top 9 gene ontology term analysis (Ingenuity Pathway Analysis, IPA). (**G**) String (string-db.org) functional pathway analysis of proteins found in mass spectroscopy of cancer juxtacrine matrix.

**Figure 3 jcm-09-00991-f003:**
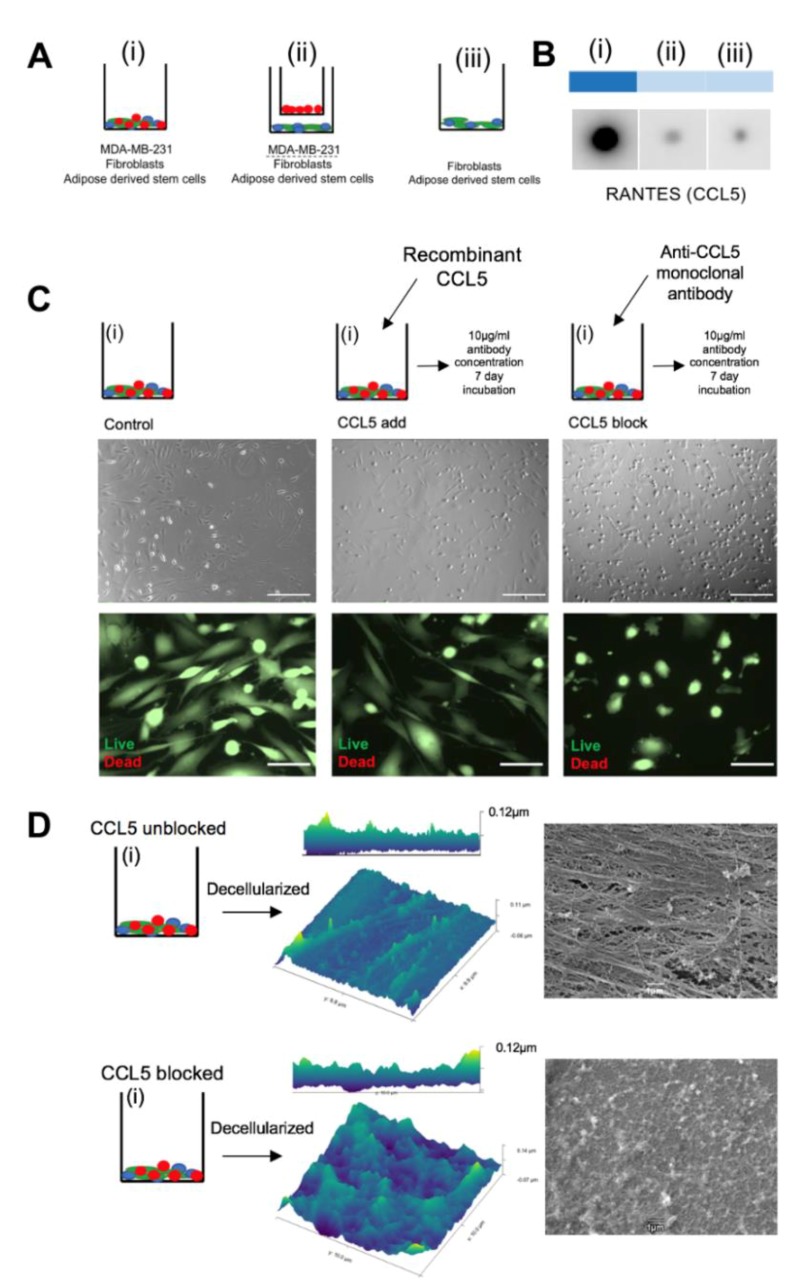
CCL5 is responsible for the linear ultrastructure of in vitro ECM (**A**) Heat map showing results from cytokine array performed on media harvested from day five of the culture period of groups (i), (ii), and (iii). (**B**) Schematic showing intervention plan of group (i) during the matrix deposition period total seven days treatment). (**C**) Top row: Brightfield images showing difference in phase contrast of cells cultured with recombinant CCL5, CCL5 monoclonal antibody, and control. Scale bar 100 μm. Bottom row: live/dead stain on the same cells shown in top row. Scale bar 50 μm. (**D**) Schematic showing cancer juxtacrine culture with was cultured normally (top) and cultured with monoclonal antibody against CCL5 (bottom). Schematic leads to AFM images of the decellularized protein from both conditions. Right hand side shows SEM images of the same matrices.

**Figure 4 jcm-09-00991-f004:**
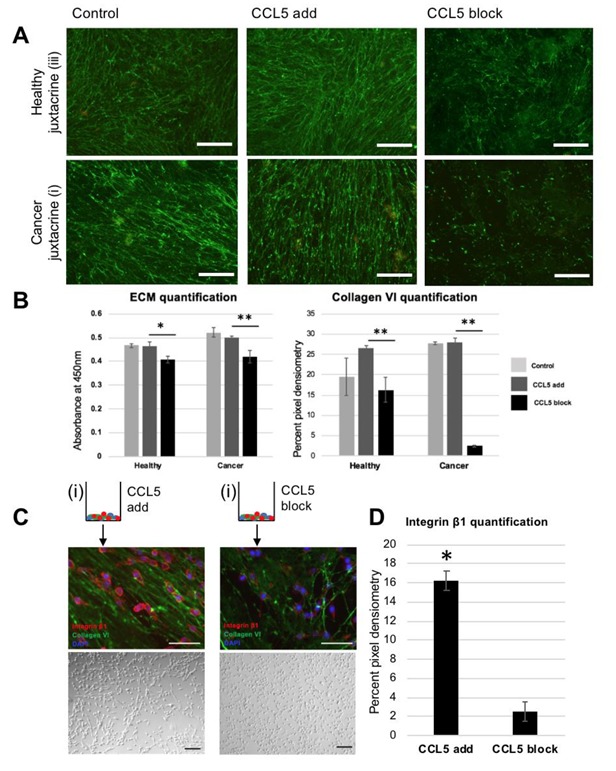
Blocking CCL5 pathway in vitro reduces both linear ultrastructure of ECM and oncogenic cell response of reseeded cells: (**A**) fluorescent images of collagen VI immunohistochemistry. Top row is staining of healthy juxtacrine (group (iii)), bottom row is staining of cancer juxtacrine (i). Left hand side shows control group staining (no dose), middle images show staining of ECM deposited while supplied with recombinant CCL5, and right-hand side shows staining of ECM deposited while supplied with monoclonal antibody against CCL5. Scale bar 100 μm. (**B**) Left: ECM quantification of matrices shown in (**A**). Right: quantification of pixel density within images in (**A**). (* *p* <0.05, ** *p* <0.01). Color legend: Light gray = control, middle grey = CCL5 added, black = CCL5 blocked. (**C**) Schematic of two treatment groups. Left side is cancer juxtacrine matrix (group (i)) schematic (with recombinant CCL5 added while matrix was being deposited), showing via immunohistochemistry reseeded MDA-MB-231 on decellularized matrix. Below is a Brightfield image of the same cells. Right side shows the cancer juxtacrine matrix (group (i)) schematic (with monoclonal antibody against CCL5 added while matrix was being deposited), showing via immunohistochemistry reseeded MDA-MB-231 on decellularized matrix. (**D**) Quantification of integrin β1 pixel density in micrographs of (**C**). (* *p* <0.05). Schematic of cancer juxtacrine group represents the cell combination forming the border analogue. Zoomed in panel shows via SEM the linearized matrix, which is found at the tumor boundary, and was produced under high levels of CCL5.

**Figure 5 jcm-09-00991-f005:**
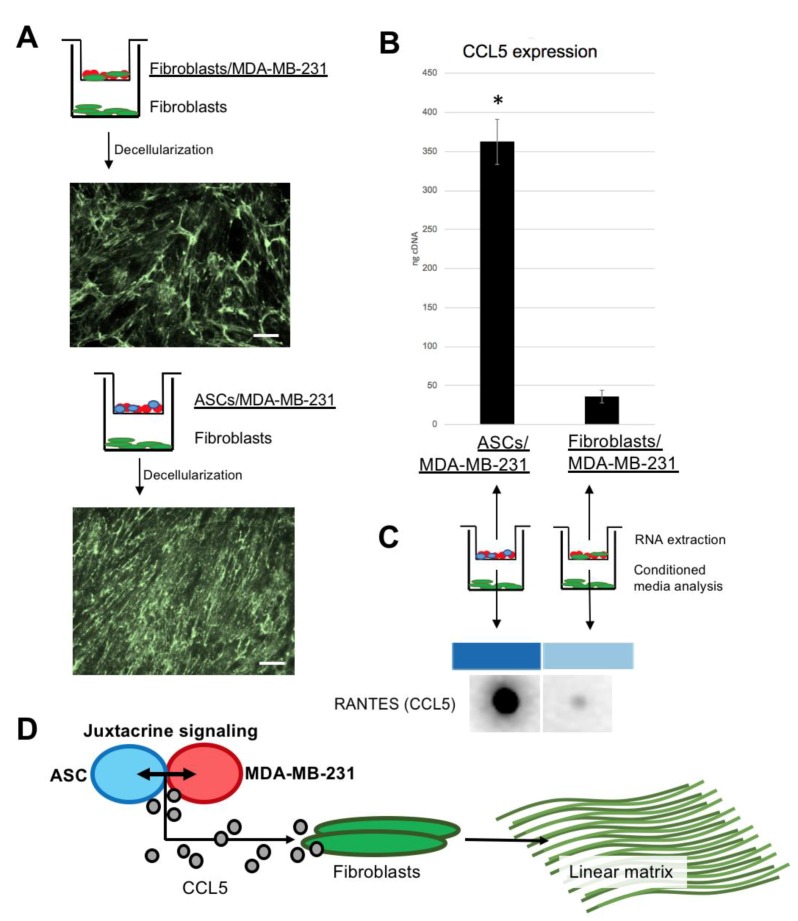
Fibroblasts responding to CCL5 from the paracrine culture of ASC/MDA-MB-231 create linear collagen type VI: (**A**) schematics and immunohistochemistry micrographs of Boyden chamber co-cultures. Top version shows fibroblasts/MDA-MB-231 in Boyden chamber, with fibroblasts beneath. Image below is immunohistochemistry for decellularized collagen VI (meshwork pattern). Bottom version shows ASC/MDA-MB-231, with fibroblasts beneath. Image below is immunohistochemistry for decellularized collagen VI (linear pattern). Scale bar 100 µm. (**B**) Graph showing qPCR data from cells cultured in Boyden chambers from (**B**). Error bars represent SD. (* *p* <0.05). (**C**) Results from blot against CCL5 of conditioned media taken from the main well of the Boyden co-culture. (**D**) Schematic illustrating hypothesized axis existing between three key cell types of invasive breast cancer; juxtacrine signaling between ASC and MDA-MB-231, which together produce CCL5, which as a stimulant to the resident fibroblasts for linear collagen VI production.

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
