# Peer review of "Oncogenic Linear Collagen VI of Invasive Breast Cancer Is Induced by CCL5"

_jcm, 2020, doi:10.3390/jcm9040991_

Round 1
Reviewer 1 Report
This paper is studying the collagen deposit of collagen produced by TNBC. The author's main discovery is that cell line of TNBC produces oncogenic collagen under the influence of CCR5.
This novel finding does add merit to this study, however there are many drawbacks:
1) There is no (as the authors point) out any in vivo correlative studies
2) There really are not any oncogenic potential studies in vitro, e.g. EMT, invasion patterns, cell proliferation etc
3) there is a lot of a=b=c and thus speculation, for instance the supposition that the HIV drug Maraviroc could be used to block CCR5, which is brought up in the results, I assumed then they would have studied it, or that they see increased integrin b1 and again bring up data in the results of other studies and other such examples
4) the addition of the ASCs was somewhat confusing and disjuncted, why is this important to even study, clearly a ratio such stem cells with fibroblasts and breast cancer would not occur in patients? Yet this was the condition its seems to get the linear collagen
If these critiques can be addressed as well as a bit more flow to the results section making it more obvious why they went from step 1 to step 2, etc this may be a paper to be considered for publication.
Author Response
- We thank the reviewer for this note. Considering that this set of revisions has a time constraint of one week, the study will be resubmitted to the journal without an in vivo
- We thank the reviewers for this perspective. We judged that the road formation phenomenon was sufficient to show a cell organization effect in this study, which was a side investigation to the main ECM research. Considering the time restraint, there is no option for us to re-run invasion/proliferation studies, however we will keep these suggestions in mind for the next upcoming study.
- We thank the reviewer for bringing up Maraviroc, and the connection we elude to in the discussion. We view testing Maraviroc as the next step in this field of research, and we would like to test it in vivo as a next step for proof of concept. There is no chance for experimenting with Maraviroc in the time constraint from the editorial office, but we thank the reviewer for the insight and will consider examining the integrin repertoire in the next set of experiments.
- We thank the reviewer for highlighting the ratios of cells used. ASCs were identified in Supplemental Figure 1 as a critical cell type for generating linearized matrix in the co-culture. Indeed, it was the use of ASCs which allowed investigation and realization of ASC/MDA-MB-231 produced CCL5, a cornerstone of this study’s hypothesis.
Since the adiposity of breast tissue varies between patients, we settled on a 1:1:1 ratio for all three cell types in vitro, as an initial test formula. We agree; no such balance exists in vivo, however this study was to help parse the cellular relationships, the source of CCL5, and the effect on fibroblasts, rather than the ratios of cell types.
Per the final note from Reviewer 1, we have amended the results section to have a logical stepwise flow by adding the following sentences:
- Line 67: "The co-culture groups were selected with special attention to creating an analogue of the invasive tumor border, while allowing for experimental control. The pathway of this work was defined by results from unbiased experimental design, hence the explorative tone of the study.", so as to manage readership framing.
- Line 177: "Given these significant morphological differences and ability to instruct reseeded MDA-MB-231 cells, the next stage was to test the protein make-up of the matrices produced."
- Line 202: Considering the high collagen VI level in the cancer juxtacrine matrix, it was important to understand next the paracrine signaling occurring among the cells of group i compared to group ii as matrices were being deposited.
- Line 239: The next important effect of blocking CCL5 to be tested is whether it affects the proportion of collagen VI in the matrix of group i, and if there is any difference in the reseeded cellular behavior on matrix formed in CCL5low conditions.
- Line 268: The next step in this study was to understand the origin of CCL5, and the main producer of linear matrix.
Reviewer 2 Report
Overall this is an excellent manuscript that is very well done. The results are clear and convincing indicating a role for ASC / TNBC interaction via CCL5 matrix reconfiguration. I have been interested in the TNBC tumor border for many years and found this paper fascinating. There is a tremendous amount of insight gained from this in-vitro work. However that is the weakness of the study. Are we just observing an in-vitro phenomena that is independent of TNBC biology. To answer that I think an additional control would be warranted. The authors elude to that in the discussion. I think only an additional Figure 1A grouping would be needed; Fibroblast, ASC + ER/PR+ tumor cell line juxtacrine.
- Minor edit - Figure 2 legend. Part E - has "justacrine" where juxtacrine has been used throughout.
- Please provide an additional control - ER/PR + breast tumor cell line.
Author Response
- We thank the reviewer for this attention to detail, we have fixed this typing mistake.
- We thank the reviewer for this well-reasoned experimental suggestion. Since there is a time constraint imposed by the editorial office, this resubmission will not have the chance to contain new in vitro test groups. However, testing cancers of different hormonal profiles will be considered for the next study arising from this initial paper.
Round 2
Reviewer 1 Report
The authors appear to have improved their study by increasing its conciseness. The in vitro only nature of the study and lack of additional cell lines do still reduce impact. Some minor English issues persist, but do not detract from the manuscript.
However, I do believe that this paper presents novel findings and despite its limitation is worthy of publication.